

# Identifying leptospirosis hotspots in Selangor: uncovering climatic connections using remote sensing and developing a predictive model

Muhammad Akram Ab Kadir[1], Rosliza Abdul Manaf[1], Siti Aisah Mokhtar[1] and Luthffi Idzhar Ismail[2]

[1] Department of Community Health, Universiti Putra Malaysia, Serdang, Selangor, Malaysia
[2] Department of Electrical & Electronic Engineering, Universiti Putra Malaysia, Serdang, Selangor, Malaysia

Corresponding author
Rosliza Abdul Manaf,
rosliza_abmanaf@upm.edu.my

## ABSTRACT

**Background.** Leptospirosis is an endemic disease in countries with tropical climates such as South America, Southern Asia, and Southeast Asia. There has been an increase in leptospirosis incidence in Malaysia from 1.45 to 25.94 cases per 100,000 population between 2005 and 2014. With increasing incidence in Selangor, Malaysia, and frequent climate change dynamics, a study on the disease hotspot areas and their association with the hydroclimatic factors could enhance disease surveillance and public health interventions.

**Methods.** This ecological cross-sectional study utilised a geographic information system (GIS) and remote sensing techniques to analyse the spatiotemporal distribution of leptospirosis in Selangor from 2011 to 2019. Laboratory-confirmed leptospirosis cases ($n = 1,045$) were obtained from the Selangor State Health Department. Using ArcGIS Pro, spatial autocorrelation analysis (Moran's I) and Getis-Ord Gi* (hotspot analysis) was conducted to identify hotspots based on the monthly aggregated cases for each subdistrict. Satellite-derived rainfall and land surface temperature (LST) data were acquired from NASA's Giovanni EarthData website and processed into monthly averages. These data were integrated into ArcGIS Pro as thematic layers. Machine learning algorithms, including support vector machine (SVM), Random Forest (RF), and light gradient boosting machine (LGBM) were employed to develop predictive models for leptospirosis hotspot areas. Model performance was then evaluated using cross-validation and metrics such as accuracy, precision, sensitivity, and F1-score.

**Results.** Moran's I analysis revealed a primarily random distribution of cases across Selangor, with only 20 out of 103 observed having a clustered distribution. Meanwhile, hotspot areas were mainly scattered in subdistricts throughout Selangor with clustering in the central region. Machine learning analysis revealed that the LGBM algorithm had the best performance scores compared to having a cross-validation score of 0.61, a precision score of 0.16, and an F1-score of 0.23. The feature importance score indicated river water level and rainfall contributes most to the model.

**Conclusions.** This GIS-based study identified a primarily sporadic occurrence of leptospirosis in Selangor with minimal spatial clustering. The LGBM algorithm effectively predicted leptospirosis hotspots based on the analysed hydroclimatic factors. The integration of GIS and machine learning offers a promising framework for disease

surveillance, facilitating targeted public health interventions in areas at high risk for leptospirosis.

## INTRODUCTION

### Background

Leptospirosis is a globally significant zoonotic disease caused by the genus *Leptospira* pathogenic bacteria (*Chacko et al., 2021*). Disease outbreaks are closely linked to heavy rainfall, flooding, and hot, humid climates (*Lau et al., 2010*). It disproportionately affects impoverished communities in developing countries, particularly those in slums, agriculture, or water-based recreation activities (*Jittimanee & Wongbutdee, 2019*; *Torgerson et al., 2015*). Transmission to humans primarily occurs when broken skin or mucous membranes encounter water or soil contaminated by urine from infected animal reservoirs (*Chacko et al., 2021*; *Karpagam & Ganesh, 2020*). While many cases exhibit mild, flu-like symptoms and may not require treatment, severe leptospirosis can manifest as disease complications such as pneumonia, kidney failure, and pulmonary haemorrhage or can even be fatal (*Karpagam & Ganesh, 2020*).

Leptospirosis is endemic in tropical and subtropical areas of South Asia, Southeast Asia, and South America. While considered a neglected disease in developed countries like the United States and Europe due to less favourable environmental conditions, outbreaks can occur *via* travel to endemic countries or engaging in water-borne activities without adequate protection (*Chacko et al., 2021*; *Karpagam & Ganesh, 2020*). The disease also flourishes in settings with poor sanitation, as these conditions support large rodent populations, a significant disease reservoir in the Southeast Asia (SEA) region (*Garba et al., 2018*). Warm tropical climates with substantial rainfall common to Southeast Asia (SEA) create environments favourable for *Leptospira* growth, promoting transmission to humans (*Nozmi et al., 2018*), with *Leptospira interrogans* and *L. borgpetersenii* identified as the key pathogenic species in the region (*Cosson et al., 2014*).

Malaysia's consistently hot and humid climate with intermittent heavy rainfall patterns creates a conducive environment for the disease, influencing leptospirosis outbreaks (*Holt, Davis & Leirs, 2006*; *Lopez et al., 2019*). The state of Selangor exhibits particularly high leptospirosis incidence, peaking in 2013 at 24.68 cases per 100,000 population (*Tan et al., 2016*). Even though the Selangor State Health Department reports the disease reduction trend in the years following, *i.e.,* 0.63 cases per 100,000 population in 2019, the actual incidence may be underestimated due to clinical overlap with other endemic diseases like dengue fever and malaria (*Benacer et al., 2016*). Furthermore, climate change may further worsen leptospirosis risk through increased flood intensity and weather fluctuations, promoting vector population growth and *Leptospira* spread (*Abdul Rahman, 2018*).

Geographic information systems (GIS) revolutionised epidemiological research by analysing and layering various spatial data to look at disease patterns (*Goodchild, 2005*). Spatio-temporal analysis of disease patterns further enhances understanding by incorporating the time dimension, facilitating understanding disease trends and the creation of predictive models for disease management (*Byun, Lee & Hwang, 2021*; *Convertino et al., 2021*). Additionally, remote sensing complements GIS in measuring the earth's surface environmental properties, such as rainfall patterns and surface temperature, by detecting electromagnetic waves (*DeMers, 2009*). These data could then be integrated into epidemiological studies to examine disease dynamics and their environmental correlations (*Dukiya, 2021*; *Tran, Kassie & Herbreteau, 2016*). Spatial data is essential for understanding disease patterns and identifying risk factors in health research. By analysing point data (*e.g.*, individual disease cases) and aggregate data (*e.g.*, regional disease rates), researchers can uncover valuable insights into the geographic distribution of diseases and potential contributing factors (*Lin & Wen, 2022*).

Spatial autocorrelation statistics (Moran's I) measure the degree of similarity between observed values at different spatial locations, with positive autocorrelation indicating an area with high values surrounded by neighbouring values with a higher incidence than other areas (*Lin & Wen, 2022*). This analysis has broad applications in identifying spatial associations of disease incidence and distribution. Studies that demonstrate the clustering of diseases at particular regions in respective researched areas include kala-azar cases in India (*Bhunia et al., 2013*), COVID-19 in Vietnam (*Thi-Bich-Thuy & Thi-Hien, 2023*), and leptospirosis in Thailand (*Chadsuthi et al., 2022*). Meanwhile, the Getis-Ord Gi* statistics identify hotspot areas by measuring the intensity of high values. Hotspot analysis reveals spatial clusters and provides visual insights into disease trends. Identifying hotspots enables targeted resource allocation, interventions, and public awareness campaigns to alleviate environmental risk factors associated with infectious diseases. Additionally, analysis of climatic variables and their association with leptospirosis hotspots could address the issues of climate change's impact on leptospirosis vulnerability, with the associated secondary chronic disease complications and economic loss from uncontrolled leptospirosis outbreaks.

Machine learning (ML) is a field within computer science that enables computers to learn and adapt without being explicitly programmed. It utilises algorithms to analyse data, identify patterns, and train models that automate decision-making processes (*Kufel et al., 2023*). Some examples of ML algorithms include artificial neural networks (ANN), support vector machine (SVM), decision tree, and Random Forest. Successful development of a hotspot predictive model relies on integrating pre-processing techniques to ensure high-quality data. Techniques like normalisation and sampling adjustments are essential in ML to address biases and data imbalances (*Abas et al., 2024*). Underreporting or mishandling geographic of confirmed cases can be handled by oversampling, where minority classes are augmented, and weights are assigned to equalise data for feature target class, ensuring a quality data input for building a robust and reliable predictive model (*Abas et al., 2024*).

## Objectives

This research aims to explore the spatio-temporal distribution of leptospirosis hotspot areas and their association with climatic factors to develop a predictive hotspot area using GIS and remote sensing methods. The study hypothesis posits a significant correlation between leptospirosis hotspot areas and specific climatic factors. Therefore, this study could (1) describe the characteristics of leptospirosis cases in Selangor from 2011 to 2019 through descriptive statistics; (2) determine the spatiotemporal distribution map of leptospirosis cases; (3) determine leptospirosis hotspot maps at the district level in Selangor; (4) describe Selangor's climatic characteristics using remote sensing imagery data (monthly rainfall and monthly land surface temperature); (5) determine the association between climatic factors and leptospirosis hotspot areas; and (6) to validate the leptospirosis hotspot area predictive model (machine learning algorithm). This life-threatening condition demands proactive hotspot prediction and outbreak management. The methodology used in this study can be adapted for future research on leptospirosis in other regions, contributing to a better understanding of disease patterns and risk factors.

## MATERIALS & METHODS

This study builds upon the authors' previously published research protocol (*Ab Kadir et al., 2023*), which outlined the methodological framework for investigating the spatiotemporal distribution of leptospirosis in Selangor. Here, we highlight further details of the methodology used in the study and the research results.

### Study design and data collection

The study was conducted in Selangor, situated in the centre of Peninsular Malaysia, bordering the Perak, Pahang, and Negeri Sembilan. It is a retrospective ecological observational study with GIS and remote sensing mapping and analysis concerning leptospirosis in Selangor using secondary data over nine years from January 2011 to December 2019. This study obtained data from various available spatial data sets, including the leptospirosis case reports, satellite images, river hydrometric levels, and topographical data of Selangor. All of Selangor's subdistrict polygon areas represent the sampling unit with a sampling size of fifty-five subdistrict polygon areas. The study utilises the universal sampling method of all notified laboratory-confirmed leptospirosis cases. Researchers examined the data to determine the sociodemographic characteristics of cases and coordinates of possible infection sources. Before the study commenced, ethical approval (NMRR ID-22-01548-C0Z IIR) was obtained from the Medical Research Committee on the Ministry of Health, the Director General of the Ministry of Health, Malaysia, and the Ethic Committee for Research Involving Human Subjects, Universiti Putra Malaysia and registered with the National Medical Research Registry.

### Operational definition of variables

Dependent variable: A leptospirosis hotspot area in this study refers to a spatially or geographically concentrated area in Selangor with a statistically significant clustering of reported cases in a subdistrict compared to surrounding subdistrict areas identified using
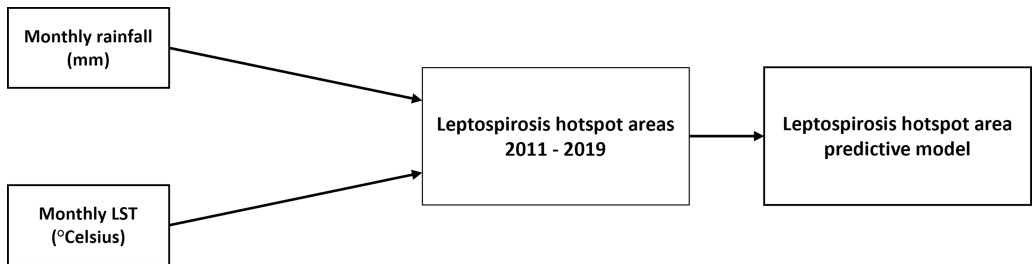

**Figure 1** **Monthly rainfall and land surface temperature (LST) contributes to leptospirosis hotspot areas. The association between the two variables were analysed using machine learning analysis.** Conceptual framework of the study.

the Getis-Ord Gi* statistics in the ArcGIS Pro software (*Esri, 2021*). Some researchers suggest including modifiers such as "transmission hotspots," "emergence hotspots" or "burden hotspots" to further explain how hotspots are defined. The burden hotspot, which is an area with elevate disease incidence or a geographic cluster of cases, describes hotspot areas in this study (*Lessler et al., 2017*). The Getis-Ord Gi* analysis examines a hotspot according to the high aggregated values (cases) for a subdistrict surrounded by subdistrict(s) with high values. In this research, the confidence levels corresponding to the binned *p*-values are labelled 'Gi_bin.' The confidence levels involve a range of values from −3 to 3. Within this range, values from −3 to −1 are categorised as "cold spot" areas, values from −1 to 1 are considered "not significant," and values from 1 to 3 are identified as "hot spot" areas. Independent variables: (1) the rainfall images in millimetres (mm) captured by the satellite will be analysed to obtain the average monthly rainfall, and (2) the LST is the earth's surface temperature captured by the satellite in degrees Celsius. Figure 1 shows the conceptual framework of this study (*NASA, 2022*). The framework outlines the geospatial processes of satellite images and their association with leptospirosis hotspot areas in Selangor from 2011 to 2019.

## Data processing and analysis

Processing leptospirosis data: The data cases were cleaned and geocoded according to their longitude and latitudes of possible infection locations. Next, the coordinates were imported to the ArcGIS Pro software and plotted as point-shape files on the layered base map to determine the spatial distribution of the leptospirosis case. Processing satellite data: Images obtained were processed to obtain the average monthly data for rainfall and LST for each subdistrict using the clipping and zonal statistic tools in the ArcGIS Pro software. The images were resampled to smaller pixels to be projected and overlayed on all subdistrict polygon shapefile boundaries. The output obtained in numerical values was exported as an Excel table for further processing and analysis. The descriptive statistics of leptospirosis cases for each month from 2011 to 2019 were analysed using Statistical Package for the Social Sciences (SPSS) software version 27.0. Descriptive data was presented using the appropriate frequency and percentage tables.

The spatio-temporal analysis of patterns and distribution of leptospirosis hotspot areas in Selangor was analysed using Moran's I and Getis-Ord Gi spatial statistical tools in ArcGIS Pro software. Moran's I analysis measures the global spatial autocorrelation between locations based on their characteristics. The tool calculates Moran's index value with a z-score and *p*-value to evaluate the significance of the index. The value usually ranges from −1 to +, and a statistically significant *p*-value and positive z-score denote that the data is more spatially clustered. The pattern derived from the analysis indicates whether the cases are clustered, dispersed, or randomly distributed. Meanwhile, the Getis-Ord Gi* statistics perform hotspot analysis of the aggregated plotted cases in a subdistrict, indicating whether it is a hot or cold spot based on the clustering and z-scores derived. It involves comparing the aggregated values (total cases for a particular subdistrict polygon area in a specific month and year) with the surrounding values. The method computes a z-score and a *p*-value to assess the spatial statistical significance of the local clusters (*Esri, 2021*).

To assess the relationship between leptospirosis hotspot areas and climatic variables, the Spearmann's correlation analysis was employed. Positive significant correlation at *p*-value <0.05 between the variables examined were examined. Subsequently, the predictive models for leptospirosis hotspot areas were developed using Python's machine learning capabilities within the Jupyter Notebook environment (Anaconda Navigator). Three well-established machine learning algorithms were selected: support vector machine (SVM), Random Forest (RF), and light gradient boosting machine (LGBM). These algorithms have been successfully applied in various leptospirosis studies to predict disease outbreaks (*Ahangarcani et al., 2019*; *Douchet et al., 2022*; *Jayaramu et al., 2023*; *Mohammadinia et al., 2019*). The best model was determined by evaluating their performance metrics in identifying actual positive hotspot areas (*Abas et al., 2024*).

# RESULTS

## Characteristics of leptospirosis cases in Selangor

The characteristics of leptospirosis cases are shown in Table 1. Between 2011 and 2019, Selangor reported 1,045 confirmed leptospirosis cases, primarily affecting males (73%) with an average age of 31. Malays were the most common ethnic group (67%), followed by Indians (11%) and Chinese (5%). Foreigners, mainly from Indonesia and Bangladesh, accounted for 14% of cases. While most patients recovered, 5% unfortunately died. Hulu Langat, Hulu Selangor, and Petaling districts had the highest number of cases during this period. The overall incidence showed a general decline with fluctuations in some districts. A peak occurred in early 2011–2012, followed by a decline until 2014. Another peak emerged in 2014 before a steady decrease until 2019.

A high-density pattern could be observed in Ampang, Klang, Kajang, and Damansara subdistricts in the Hulu Langat, Petaling, Gombak, and Klang districts, with cumulative cases ranging from 58 to 109. Cases were more concentrated in Selangor's central and northeastern regions, while other subdistricts show a more scattered distribution. Throughout 108 months from 2011 to 2019, case distribution was discovered in 103 months, mostly showing a random distribution pattern with the clustering of cases

**Table 1** Sociodemographic characteristics of leptospirosis cases ($n = 1,045$) in Selangor (2011–2019) based on available data retrieved from the Selangor State Health Department. Characteristics of leptospirosis cases in Selangor ($n = 1,045$).

| Characteristics | Mean (SD) | Frequency, $n$ | Percentage (%) |
|---|---|---|---|
| **Gender** | | | |
| Male | | 765 | 73.01 |
| Female | | 282 | 26.99 |
| **Age** | | | |
| | 31 (18) | | |
| **Race** | | | |
| Malay | | 703 | 67.27 |
| Foreigners | | 142 | 13.59 |
| Indian | | 115 | 11.00 |
| Chinese | | 57 | 5.45 |
| Pribumi Sabah/Sarawak | | 11 | 1.05 |
| Orang Asli | | 9 | 0.86 |
| Others | | 8 | 0.77 |
| **Patient's status** | | | |
| Alive | | 995 | 95.22 |
| Dead | | 50 | 4.78 |

occurring in certain months. Case clustering was observed in Selangor's central regions in the earlier months of 2011 before shifting toward the western coastal areas of the state. Cases were mostly randomly distributed throughout the state in 2012, while clustering was observed in the third quarter of 2013 and 2014. The distribution of cases declined from 2015 to 2019, with yearly cases ranging from 41 to 78 cases. Even though case plotting shows a random pattern along the years, clustering of cases was observed primarily in months of the first and fourth quarters of the years.

## Spatio-temporal analysis

Moran's I analysis confirmed spatial clustering of cases in 20 out of 103 months, with a positive z-value (0.067 to 0.370) and statistically significant $p$-values. Meanwhile, a dispersed distribution of cases was observed in June 2018, giving a negative Moran's I index z-value. The Getis-Ord Gi* analysis depicts statistically significant hotspots where subdistricts with similar high cases surround subdistricts with higher leptospirosis cases.

The spatial relationship was conceptualised at a fixed distance band, in which each feature is analysed within the context of neighbouring features and receives a weightage according to the specified threshold distance. The Euclidean distance method, which is the straight line distance of subdistrict polygons containing aggregated leptospirosis cases, was used in the analysis. The hotspot maps areas throughout the months were mostly scattered in subdistricts throughout Selangor, with clustering mainly observed in the central regions in Selangor. The supplementary figures provide further insights into monthly hotspot areas following Getis-Ord Gi* analysis.
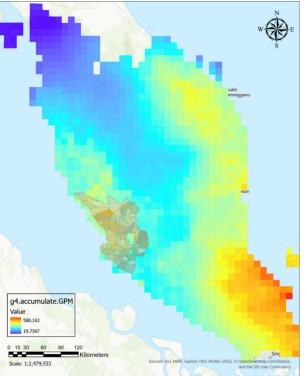 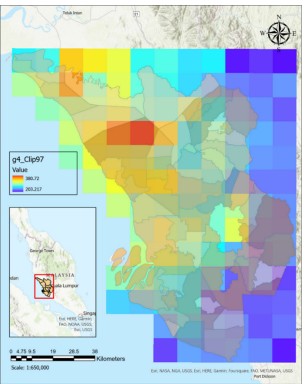 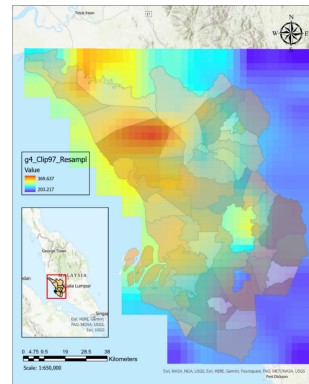

**Figure 2** **Satellite data processing for downloaded rainfall raster data for Malaysia from clipping to Selangor boundaries to data extraction for monthly average rainfall data for every subdistrict polygon shapefile.** (A) Processed satellite image for rainfall raster data in Malaysia. (B) Rainfall raster data clipped with Selangor boundaries. (C) Resampled rainfall raster data for Selangor.

**Table 2** **Bivariate correlation analysis using Spearman's rho statistics between leptospirosis hotspot area and climatic factors analysed in the study.** Correlation between leptospirosis hotspot area and climatic factors in Selangor.

| Variable | | Median (IQR) | n (%) | r | p-value |
|---|---|---|---|---|---|
| **Monthly rainfall** | | | | −0.006[a] | 0.627 |
| Minimum (mm) | 3.62 | 201.22 | | | |
| Maximum (mm) | 613.83 | (139.78) | | | |
| **Monthly LST** | | | | −0.086[a] | <0.001 |
| Minimum (°C) | 18.67 | 29.64 | | | |
| Maximum (°C) | 40.89 | (3.24) | | | |

**Notes.**
*p-value less than 0.05.
[a] Spearman's correlation coefficient.

This study employed statistical methods to assess the link between climate factors (monthly rainfall and land surface temperature) and areas identified as leptospirosis hotspot areas. Figure 2 shows satellite data processing steps for a processed rainfall image for a particular month. The satellite image was resampled to appropriate pixels to fit the subdistrict boundaries before zonal statistics were performed to extract monthly average rainfall and LST values for each subdistrict. The similar process was repeated for all months throughout the study period.

The analysis in Table 2 revealed a statistically significant but weak, negative correlation only between monthly land surface temperature and leptospirosis hotspots (correlation coefficient, $r = -0.086$, p-value < 0.001). Conversely, no significant correlations were found for monthly rainfall ($r = -0.006$, p-value = 0.627). These weak associations in the initial analysis suggest that a more sophisticated approach might be necessary to capture the complexities underlying these relationships. To address the complexity of these relationships, we employed machine learning techniques, as supported by literature.

**Table 3  Model performance generated using machine learning analysis.**

| Model | LGBM | Random forest | SVM |
|---|---|---|---|
| Cross-validation score | 0.61 | 0.61 | 0.62 |
| Test Score | 0.62 | 0.62 | 0.61 |
| Precision | 0.16 | 0.13 | 0.14 |
| Sensitivity | 0.43 | 0.50 | 0.41 |
| F1-score | 0.23 | 0.20 | 0.21 |

## Predictive model

After the hyperparameters of the aforementioned algorithms were optimised, a split dataset was used to evaluate each model's performance. Using a variety of indicators, we assessed the algorithms' ability to predict leptospirosis hotspots. These included cross-validation scores on the training data and the unseen hold-out dataset with the trained algorithm. Furthermore, sensitivity, accuracy, and F1-score were assessed to choose the best model for hotspot prediction.

Based on the analysis, the best model for predicting leptospirosis hotspot areas was the LGBM algorithm. Compared to RF (0.13) and SVM (0.14), LGBM achieved the highest precision (0.16), suggesting a more robust capacity to categorise actual hotspot locations correctly. When identifying all hotspots, LGBM did not show the highest value for sensitivity (0.43) (RF: 0.50, SVM: 0.41). However, LGBM obtained the greatest F1-score (0.23), balancing sensitivity and precision. This can be inferred that LGBM finds the best possible balance between minimising false positives, predicting locations as hotspots when they are not, and finding actual hotspots. Table 3 provides a thorough analysis of each model's performance comparison.

The most significant factors influencing the model's predictions can be identified through feature importance analysis. Each input variable is scored according to how well it predicts the target variable (hotspot areas) in the model. A higher score denotes a more substantial influence on hotspot prediction. Notably, only the LGBM and RF algorithms allow feature-importance computation. Rainfall was found to be the most important factor by LGBM, followed by LST (Fig. 3). On the other hand, RF showed the opposite significance hierarchy.

## DISCUSSION

This study examined 1,045 laboratory-confirmed leptospirosis cases over nine years. Consistent with prior research, males comprised the majority of cases (73%), likely due to their increased exposure to contaminated environments through outdoor activities (*Ko et al., 1999*; *Naing et al., 2019*). Similarly, a global review of outbreaks from 1970 to 2012 found that males comprised two-thirds of cases, highlighting their higher risk during recreational activities (*Munoz-Zanzi et al., 2020*). The study also found Malays, the dominant ethnic group in the region, to be the most affected population (67%), which likely reflects the broader population demographics rather than specific susceptibility.

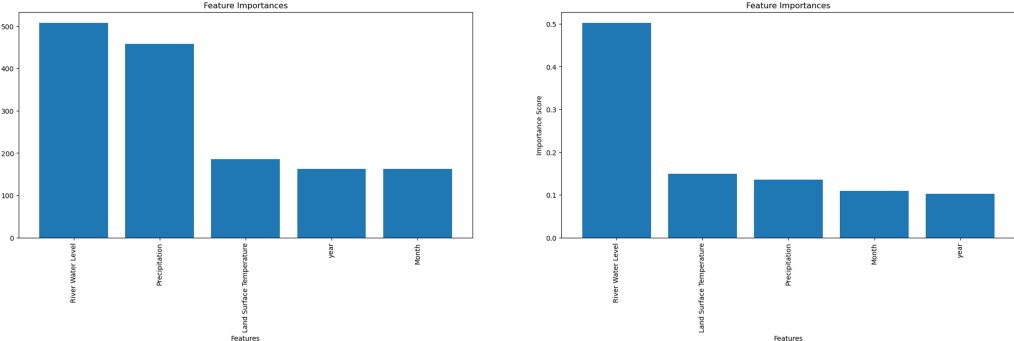

**Figure 3** **Machine learning feature importance analysis can only be performed with the light gradient boosting machine (LGBM) and Random Forest (RF) algorithms.** (A) Feature importance for LGBM algorithm. (B) Feature importance for Random Forest algorithm.

Furthermore, the study identified younger adults as the most affected age group, a crucial demographic for national productivity (*Torgerson et al., 2015*). This suggests potential socioeconomic factors or cultural practices influencing their exposure risks. A separate study by *Nozmi et al. (2018)* found that despite good general knowledge about leptospirosis, poor preventive practices and healthcare-seeking behaviour may contribute to their vulnerability during outbreaks. An encouraging trend observed in the study was the decline in leptospirosis incidence and mortality rates over the nine years. This decrease could be attributed to improved public awareness and educational efforts regarding leptospirosis. The behaviour shift to better preventive practices could have reduced the exposure risks (*Zhang et al., 2023*). Additionally, implementing the Ministry of Health's official Leptospirosis Management Guidelines in 2011 has likely enhanced diagnosis, treatment protocols, and overall disease management. However, it is crucial to acknowledge that underreporting of cases remains a possibility, and continued surveillance is necessary to ensure sustained progress in combating leptospirosis (*Jittimanee & Wongbutdee, 2019*).

Through spatial analysis, researchers can map disease outbreaks and investigate the potential influence of environmental factors, such as climate or meteorological phenomena, on disease patterns. Combining epidemiological and geographic data provides a more comprehensive understanding of the disease's complex dynamics, ultimately assisting in effective control and prevention strategies (*Lau et al., 2010*; *Mwachui et al., 2015*). The wet season (November–March) and dry seasons (May–June) might influence human behaviour by engaging in events related to heavy rainfall, such as floods, or water related-activities in the latter, for cooler environments. This study's findings suggest that local factors rather than uniform spatial trends drive sporadic outbreaks. Investigating these local factors, especially environmental variables during clustering periods, could offer crucial insights into outbreak triggers.

Hot Spot analysis using the Getis-Ord Gi* statistic identifies hotspot areas by examining the spatial aggregation of cases within neighbouring subdistricts (*Esri, 2021*), allowing for targeted and data-driven interventions that are more informative for infectious disease management than simply knowing the location of case clusters. Identifying

leptospirosis hotspots emphasises the necessity of understanding the local environmental and socioeconomic factors that influence disease dynamics. While inadequate drainage systems, poor sanitation infrastructure, and occupational exposure to contaminated environments have been identified as risk factors for leptospirosis infection (*Lau et al., 2010*), investigating the relationship between these hotspots and climatic variables would provide an additional valuable perspective. A more sustainable approach could be to design targeted interventions that address the underlying causes of disease persistence rather than relying solely on reactive actions during outbreaks (*Allen et al., 2017*).

The random distribution of leptospirosis cases found in Selangor contrasts with a study conducted in a neighbouring country, Thailand, by *Chadsuthi et al. (2022)* where case distributions showed a more clustered pattern. However, a similar finding was observed in a study conducted in Sarawak (*Kira et al., 2022*), where spatial autocorrelation of the cases showed mixed random and clustered patterns. This similarity could be attributed to the relatively consistent climate between Selangor and Sarawak and perhaps other shared risk factors within the context of Malaysian culture. Despite these factors, Hot Spot analysis using the Getis-Ord Gi* statistic identifies hotspot areas by examining the spatial aggregation of cases within neighbouring subdistricts (*Esri, 2021*), allowing for targeted and data-driven interventions that are more informative for infectious disease management than simply knowing the location of case clusters.

The temporal analysis performed in this study showed another feature of leptospirosis epidemiology, suggesting that specific months were associated with increased illness incidence (*Mao et al., 2019*). The disease's cyclical structure suggests a possible link with seasonal variations in climatic factors such as rainfall, humidity, and temperature. These factors also alter river water levels, in which a potential overflow contributes to the spread of *Leptospira* to humans *via* contaminated water or soil (*Lopez et al., 2019*). Understanding the temporal dynamics of disease transmission allows public health officials to make more informed decisions about outbreak preparedness and the strategic implementation of preventive interventions. Subdistricts in a highly urbanised region with a dense population present more potential for contact with polluted urine, especially among risk populations.

This study discovered a minor relationship between leptospirosis hotspots and climatic factors. Even though it is clear that studies have found correlations between leptospirosis and environmental elements like temperature and rainfall (*Cunha et al., 2019*; *Lopez et al., 2019*), some researchers have discovered different results. It has been shown that there is increasing evidence of the prolonged survival of *Leptospira* in soil, which can be washed away to water bodies or soil, particularly in heavy rainfall, which poses a risk of infection to humans at risk. Research done in Salvador, Brazil, demonstrated that leptospirosis infection risk was inversely associated with rainfall, and it occurs throughout the year with bouts of increased infection severity during heavy rainfall periods.

In addition, *Warnasekara et al. (2022)* postulated that there is growing evidence of the prolonged survival of *Leptospira* in soil, which can be washed away to water bodies or soil, particularly in heavy rainfall, posing a risk of infection to humans at risk. Rainfall and land surface temperature (LST) are key climatic factors influencing leptospirosis transmission by affecting the survival and dispersal of *Leptospira*. River run-off from continuous heavy rain
can further influence pathogen contamination from water dispersal to surrounding lands (*Lopez et al., 2019*). Traditional statistical approaches may have limitations in capturing the complex nonlinear interactions between variables and leptospirosis hotspot areas, as observed in bivariate analysis. Machine learning algorithms, such as SVM, RF, and LGBM, are well-suited to handle these complex interactions and identify patterns that may not be apparent from traditional statistical methods. This makes them a valuable tool for investigating these relationships and developing accurate predictive models for leptospirosis hotspot areas.

Machine learning provides an alternative approach for investigating these relationships and potentially developing leptospirosis hotspot area prediction models. Constructing a reliable predictive model requires splitting the dataset into training, validation, and test sets. The training set trains the algorithm on data patterns, while a separate validation set is used to assess the model's performance and prevent overfitting. Overfitting occurs when a model memorises specific training data patterns, leading to poor performance on unseen data. To overcome this, the study employed a stratified 10-fold cross-validation. This technique iteratively splits the data into ten folds, using nine folds for training in each iteration and the remaining fold for validation. This approach ensures a more robust model with improved generalizability to unseen datasets (*Baheti, 2021*). The LGBM algorithm demonstrated superior precision (0.35) compared to the RF (0.31) and SVM (0.20). Although LGBM showed slightly lower sensitivity (0.53) than the other models, it achieved the highest F1-score (0.42), indicating a superior balance between precision and recall. This suggests that the LGBM algorithm was able to identify the most true positives (hotspot area) while minimising false positives, outperforming the other two algorithms (*Jagarlapoodi, 2023*).

This study compared the performance of SVM and RF algorithms for predicting leptospirosis hotspots. SVM excels at identifying linear separations between classes but requires data transformation for non-linear problems (*Ben-Hur & Weston, 2010*). While SVMs have achieved high accuracy in disease prediction in some studies (*Douchet et al., 2022*; *Kim & Ahn, 2021*), imbalanced data can hinder their performance (*Maldonado, Weber & Famili, 2014*). RF, conversely, performed well with an accuracy of 82.6% and identified rainfall as the key factor, aligning with our findings (*Jayaramu et al., 2023*). However, imbalanced data can still pose challenges, needing modifications like Weighted Random Forest and Balanced Random Forest methods to be initially applied (*Chen & Breiman, 2004*). Notably, the LGBM outperformed both algorithms with a higher precision score, demonstrating its effectiveness in handling imbalanced data and making it the best choice for identifying true leptospirosis hotspots in this study.

To increase computation speeds and accuracy when working with huge datasets, the LGBM, an enhanced gradient boosting decision tree algorithm, applies the gradient-based one-side sampling (GOSS) and exclusive feature bundling (EFB) techniques (*Ke et al., 2017*). After adjusting the specified hyperparameters, the method builds a robust model using the histogram approach, which further derives from learning from the decisions of previous trees (*Khadka, 2024*). It would, therefore, be an innovative approach to infectious disease prediction approaches, enhancing preparedness for potential disease outbreaks with hotspot area prediction.

Limited research exists on applying machine learning algorithms for leptospirosis prediction. Existing studies, like those conducted in Seremban and Kelantan, explored models such as exploratory data analysis with artificial neural networks (EDA ANN) and RF (*Jayaramu et al., 2023*; *Rahmat et al., 2020*). While integrating these models into disease surveillance systems remains an open challenge, a multidisciplinary collaboration involving public health officials, data scientists, engineers and software developers will be crucial for bridging this gap (*Bertagnolli, 2023*). This study contributes to this evolving field by demonstrating the effectiveness of the LGBM algorithm for leptospirosis hotspot area prediction. Further research can explore the development of user-friendly interfaces and real-time data integration for seamless integration with existing disease surveillance systems.

## CONCLUSIONS

The study's spatiotemporal distribution of leptospirosis in Selangor was visualised using GIS. The results showed a largely sporadic pattern with minimal clusters. This visualisation tool can be helpful for public health authorities as it gives a clear picture of the typical locations of incidents, enabling them to make focused interventions. Moreover, the LGBM algorithm with machine learning could detect leptospirosis hotspots accurately, demonstrating how artificial intelligence (AI) can transform disease surveillance by highlighting high-risk regions. The predictive capabilities enable authorities to forecast outbreaks and focus preventive efforts in the most vulnerable locations.

This research expands our understanding of leptospirosis epidemiology in Selangor by leveraging GIS software for hotspot analysis, data interpretation, and visualisation. It could empower public health practitioners to identify disease distribution patterns and plan targeted outbreak mitigation strategies at the regional level, ultimately contributing to global efforts against this re-emerging disease. Furthermore, our application of machine learning, particularly the LGBM algorithm for hotspot prediction, represents an innovative approach with broader implications. This demonstrates the potential of data-driven strategies to optimise traditional disease surveillance practices. While links between hotspots and climatic factors were established, the study focused on only two factors. Future research should incorporate a broader range of environmental variables, such as river water level, soil characteristics, humidity, and proximity to water bodies, to refine predictive models.

### Funding
The Putra Focus Grant (No. 6302013-14001) funded this research. The funders had no role in study design, data collection and analysis, decision to publish, or preparation of the manuscript.

### Grant Disclosures
The following grant information was disclosed by the authors:

The Putra Focus Grant: No. 6302013-14001.

## Competing Interests

The authors declare there are no competing interests.

## Author Contributions

- Muhammad Akram Ab Kadir conceived and designed the experiments, performed the experiments, analyzed the data, prepared figures and/or tables, authored or reviewed drafts of the article, and approved the final draft.
- Rosliza Abdul Manaf conceived and designed the experiments, prepared figures and/or tables, authored or reviewed drafts of the article, and approved the final draft.
- Siti Aisah Mokhtar performed the experiments, authored or reviewed drafts of the article, and approved the final draft.
- Luthffi Idzhar Ismail performed the experiments, authored or reviewed drafts of the article, and approved the final draft.

## Ethics

The following information was supplied relating to ethical approvals (*i.e.,* approving body and any reference numbers):

Ethical approval to conduct the research was obtained from the Medical Research Committee, Ministry of Health, Malaysia, and the Universiti Putra Malaysia Ethics Committee, Jawatankuasa Etika Universiti untuk Penyelidikan Melibatkan Manusia (JKEUPM). This research was registered with the National Medical Research Registry NMRR ID-22-01548-C0Z (IIR) on 25 August 2022 and post-ethical renewal on 26 September 2023.

## Data Availability

The raw data are available in the Supplemental Files.

## Supplemental Information

Supplemental information for this article can be found online at http://dx.doi.org/10.7717/peerj.18851#supplemental-information.

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
