# Peer review of "Identifying leptospirosis hotspots in Selangor: uncovering climatic connections using remote sensing and developing a predictive model"

_PeerJ, doi:10.7717/peerj.18851_

## Round 0.1 · original submission · Major Revisions

Dear Dr. Kadir and colleagues:

Thanks for submitting your manuscript to PeerJ. I have now received two independent reviews of your work, and as you will see, the reviewers raised some concerns about the research. Despite this, these reviewers are optimistic about your work and the potential impact it will have on research studying leptospirosis outbreaks. Thus, I encourage you to revise your manuscript, accordingly, considering all the concerns raised by both reviewers.

Please revise your manuscript for clarity and limit jargon/reduce verbiage but be clear about the focus of the study and the target audience. Focus on especially on clarity, and address the sections considered incomplete and/or unclear by the reviewers. It appears that certain key references are missing. The Methods should be clear, concise and repeatable. Please ensure this. Also, elaborate on the discussion of your findings, placing them within a broad and inclusive body of work by the field. Please supply any code or scripts, model explanations, etc. in the supplemental material.

I look forward to seeing your revision, and thanks again for submitting your work to PeerJ.

Good luck with your revision,

-joe

Reviewer 1 ·

Basic reporting

SPECIFIC COMMENTS
The paper is a a potentially interesting application of a set of models for Leptospirosis but I am wondering what is really new in terms of basic research (of Leptospirosis and/or methods) or applications...
Very importantly, do you predict the number of cases over time OR the total cases OR just areas as hotspots (I guess proportional to the number of cases?). It is not clear and you should really show visually how model predictions match data. Of course predicting hotspots is quite easy than predicting number of cases (say weekly) over time OR the total cases.
Some specific comments below...

-- how is Malaysia and particularly Selangor special for understanding Leptospirosis wolrdwide in more general terms considering its basic ecology and controls?
-- ''2005 and 2014. This rise coincides with the observed dynamics of climate change''... how can you say that this is a period of climate change higher than others?
-- ''Bivariate analysis indicated a weak but statistically signifcant negative correlation (r = -0.086) between LST and the occurrence of leptospirosis hotspots. '' correlation does not imply causation ... and the sign of correlation may not be meaningful because important time delays may occur (see Convertino et al. 2021)
--Fig. 1 is just YOUR MODEL and not how leptospirosis works. Any model is an interpretation of the reality based on the questions you ask or your beliefs. So, predictions are just a mental models' outputs, so a transformation of inputs into something that is a combination of inputsXprevious knowledgeXperceptions.
--Fig. 2. the resolution is low
--Fig.3. why not to use the original resolution if data are at higher resolution??
--Fig. 4. See my comments below about variables' interactions...

Experimental design

GENERAL COMMENTS:
(1) The ecological-environmental processes (Leptospirosis) you investigate are largely non-linear. Non-linear models such as Convergent Cross Mapping or others (see Sugihara et al 2012, and Li and Convertino et al. 2021 for instance, or Servadio and Convertino, 2018 for coupling networks and systemic indicators) can account for variable non-linear interactions (vs. simple correlation) even without considering time delays. These models are also able to capture spatial (eco-environmental networks) variability to understand spatial dependencies that are important for the variables considered, as well as variable collective interaction (see Convertino et al. 2021 for Lepto). I am not sure how your study can consider these non-linearities but it would be nice because any ecological outcome (and risk) is largely determined by ecological non-linearities (beyond structure that you capture). E.g. it would be also nice to map the spatial ecological corridors (e.g, preferential directions of Rinaldo at el al (2018,2020)). You are lucky that you have time series (I think) and those are really nice to run the aforementioned models.

Validity of the findings

(2) To address the model/data Uncertainty-Sensitivity coupling, global sensitivity and uncertainty analysis (GSUA) should be done to identify key determinants of model/data indicator variability and universal determinants across geographies for Leptospirosis. The authors do not quite perform even a one-factor-at-a-time sensitivity analysis and then they are not capturing the variables' linear and non-linear interactions. See Pianosi et al. (2016) for an extensive discussion about this topic and how data should be used for GSUA using a simple variance-based approach. It is essentially looking into how much variability is contained in inputs for the variability of outputs (or even better into the co-predictability versus causality based on probability distribution functions, i.e. pdfs). This is important to identify the stability and source of uncertainty of defined ecological corridors and their services (such as mappable and unknown sources of CEC pattern variability).

(3) How indicators/predicted variables (i.e. Leptospirosis pattern features, etc) change over space and time is critical to understand site-/time-specific and universal (ecosystem invariant) shifts bure more importantly the shape of stress-response patterns (e.g. how Leptospirosis features change the risk where the risk should be a systemic risk because Leptospirosis area's risk is interdependent on spatially dependent areas). This can be related to how the ecology (broadly defined) is shaped and this can be taken into account by considering the network that imposes physical constraints on ecosystems. Thus, indicator distributions (of predicted patterns) can be analyzed as a function of the predictors (possibly basin-based) considering their joint probability distribution functions (pdfs), or average value and variance and looking into indicator variability along predictors' gradients. The stability of ecological patterns over predictors' gradients is important to be quantified because that can define potential stable states over which the predictors are relatively stable or approaching a change.

Additional comments

RECOMMENDATION:
I suggest accepting the paper after very Major Revisions. The paper is potentially good but I feel like it should be improved, particularly considering the characterization of systemic risk and key network based controls, and additionally about GSUA. The material presented is not very visually appealing and many things are missing to be visualized. Also, what is really the novelty here?



REFERENCES:
M Convertino, A Reddy, Y Liu, C Munoz-Zanzi, (2021), Eco-epidemiological scaling of Leptospirosis: Vulnerability mapping and early warning forecasts, Science of The Total Environment 799, 149102

Li J, Convertino M (2021) Temperature increase drives critical slowing down of fish ecosystems. PLoS ONE 16(10): e0246222. https://doi.org/10.1371/journal.pone.0246222

Pianosi et al. (2016)
Sensitivity analysis of environmental models: A systematic review with practical workflow
Environmental Modelling & Software
Volume 79, May 2016, Pages 214-232
- https://www.safetoolbox.info/info-and-documentation/

Rinaldo at el al (2020)
River Networks as Ecological Corridors
Species, Populations, Pathogens
https://www.cambridge.org/core/books/river-networks-as-ecological-corridors/09DF57A07BA510393F04E9FEF5F838B3

Rinaldo at el al (2018)
River networks as ecological corridors: A coherent ecohydrological
perspective, Adv in Water res
https://water.usask.ca/documents/dls-2021/dls-discussion_rinaldo.pdf

JOSEPH L. SERVADIO AND MATTEO CONVERTINO
Optimal information networks: Application for data-driven integrated health in populations
SCIENCE ADVANCES
VOL. 4, NO. 2

Sugihara G et al (2012)
Detecting Causality in Complex Ecosystems
https://www.science.org/doi/10.1126/science.1227079

Reviewer 2 ·

Basic reporting

Abstract
- Start the abstract by describing the relationship between the environmental factors chosen and leptospirosis.
- With respect to the Getis-Ord Gi* result, it would be more interesting to say where the clusters with the highest incidence.

Introduction
- This research should provide additional justification for the use of remote sensing data and include relevant references to previous studies.

Experimental design

- ML including the Support Vector Machine (SVM), Random Forest (RF), and Light Gradient Boost Machine (LGBM) : explain why the ML model (SVM, RF, and LGBM) was chosen for this study. Were there any assumptions to be considered or not? If so, were they considered? Reference this method.
- Getis-Ord Gi*: well-explained method with support references cited.
- Bivariate must be better explained, including how its result should be interpreted.
-lines 162-164 ,174-180 were not shown or references given

Validity of the findings

Results

- Lines 230-232 did not show the Leptospirosis incidence report.
-Which Figures or Table corresponds to lines 234-241,234-241, 243-248, and 271-279?
-Check the accuracy of the details in line 235.
-Where are the results of Global Moran'I and Getis-Ord Gi not commented. Please talk about it.
-When considering Figures 2-3, the results of this research show that only December. Therefore the results may be inconclusive.
-The interpretation of results and the Figures are not consistent.

Discussion

- Explain better the result of Getis-Ord Gi* rather than just saying that this area is a hotspot. For example, what does hotspot mean in a particular and the neighbour.
- lines 353-354 give cited

---

## Round 0.2 · accepted · Accept

Dear Dr. Kadir and colleagues:

Thanks for revising your manuscript based on the concerns raised by the reviewers. I now believe that your manuscript is suitable for publication. Congratulations! I look forward to seeing this work in print, and I anticipate it being an important resource for groups studying the leptospirosis outbreaks. Thanks again for choosing PeerJ to publish such important work.

Best,

-joe

Reviewer 1 ·

Basic reporting

To be frank, the authors did not reply sufficiently to my comments of the review... rather they said ''we did not do it ...'' but the paper has been modified and the content is more clear. I think this is a modeling application but not quite a paper with generalizable findings (methodologically and otherwise). Nonetheless it can be published.

Experimental design

limitation about global sensitivity and uncertainty analysis that must be a must for any study ....

Validity of the findings

specific but not generalizable